# Evaluation of radiation treatment volumes for unknown primaries of the head and neck in the era of FDG PET

**Alexis Platek[1], Michael Mix[2], Varun Chowdhry[3], Mark Farrugia[3], Michael A. Lacombe[2], Jeffrey A. Bogart[2], Luke Degraaf[1], Austin Iovoli[1], Hassan Arshad[4], Kimberly Wooten[4], Vishal Gupta[4], Wesley L. Hicks Jr[4], Mary E. Platek[5,6], Seung S. Hahn[2], Anurag K. Singh[3] ***

**1** Jacobs School of Medicine and Biomedical Sciences, University at Buffalo, The State University of New York, Buffalo, NY, United States of America, **2** Department of Radiation Oncology, Upstate Medical University, Syracuse, NY, United States of America, **3** Roswell Park Comprehensive Cancer Center, Department of Radiation Medicine, Buffalo, NY, United States of America, **4** Roswell Park Comprehensive Cancer Center, Department of Head and Neck/Plastic and Reconstructive Surgery, Buffalo, NY, United States of America, **5** Roswell Park Comprehensive Cancer Center, Department of Biostatistics, Buffalo, New York, United States of America, **6** Department of Dietetics, D'Youville College, Buffalo, New York, United States of America

* Anurag.Singh@RoswellPark.org

**Data Availability Statement:** Data cannot be shared publicly because of protected health information. Data are available from the respective center Institutional Data Access / Ethics Committee

## Abstract

### Objectives

Positron-emission tomography (PET) has improved identification of the primary tumor as well as occult nodal burden in cancer of the head and neck. Nevertheless, there are still patients where the primary tumor cannot be located. In these situations, the standard of care is comprehensive head and neck radiation therapy however it is unclear whether this is necessary. This study examines the effects of radiation treatment volume on outcomes among using data from two cancer centers in unknown primary carcinoma of the head and neck.

### Methods

Patients received unilateral (n = 34), or bilateral radiation (n = 28). Patient factors such as age, gender, smoking history, and patterns of failure were compared using Mann Whitney U and Chi Square. Overall survival (OS) and disease free survival (DFS) trends were estimated using Kaplan-Meier survival curves. Effect of treatment volume on survival was examined using multivariate cox proportional hazard regression model.

### Results

No significant differences were observed in the frequency of local (p = 0.32), regional (p = 0.50), or distant (p = 0.76) failures between unilateral and bilateral radiation therapy. By Kaplan-Meier estimates, OS (3-year OS bilateral = 71.67%, unilateral = 77.90%, p = 0.50) and DFS (3-year DFS bilateral = 77.92%, unilateral = 69.43%, p = 0.63) were similar between the two treatment approaches. Lastly, multivariate analysis did not demonstrate

(contact via email) for researchers who meet the criteria for access to confidential data. Please contact RSPAdmin@roswellpark.org regarding the head and neck database under EDR-103707.

**Funding:** The author(s) received no specific funding for this work.

**Competing interests:** The authors have declared that no competing interests exist.

any significant differences in outcome by treatment volumes (OS: HR = 0.74, 95% CI: 0.31, 1.81, p = 0.51; DFS: HR: 0.68, 95% CI: 0.24, 1.93, p = 0.47).

## Conclusions

Unilateral radiation therapy compared with bilateral produced similar survival.

## Introduction

Head and Neck cancers of unknown primary origin represent a disease in which regional spread of disease is detected when the primary site is not able to be determined after diagnostic work up [1]. Unknown primaries of the head and neck represent approximately 3% of all head and neck cancers, with the majority being squamous cell carcinoma [1,2]. The diagnostic work up for these patients typically includes a fine needle aspiration followed by clinical exam of the most likely locations for the primary tumor [2]. If clinical exam fails to identify a primary tumor, imaging studies and close physical examination, often including direct laryngoscopy with or without "blind" biopsies [3], are used to identify the site. The use of positron emission tomography (PET) scans in diagnostic work up has significantly improved identification of primary tumors, nodal involvement, and has narrowed the population of patients with unknown primary tumors of the head and neck [4,5]. However even in the PET era, there remains a subset of patients where a primary tumor cannot be identified.

Current treatment options for unknown primaries of the head and neck remain controversial and include surgery, concurrent chemoradiation (CCRT), radiation alone (RT), or surgery followed by CCRT or RT [2]. In patients treated with radiotherapy, the volume of treatment is debated. Specifically, it is unclear whether to use unilateral radiation therapy (radiation to one side of the head and neck, usually covering the ipsilateral tonsil and base of tongue) versus comprehensive radiation therapy (radiation therapy to all likely mucosal sites and both sides of the head and neck). One prospective study evaluating the use of unilateral vs. bilateral radiation therapy opened in 2002 and unfortunately was closed due to low accrual rates [2]. The majority of retrospective studies which attempt to address the question of unilateral versus bilateral irradiation include patients treated before the advent of PET scans [5] or newer radiation techniques, such as Intensity Modulated Radiation Therapy (IMRT) [2]. Many of the patients in these earlier investigations may have had their primary identified and/or better characterized bilateral neck involvement by modern imaging and as such, it is unclear how well they represent a contemporary cohort. For example, former pre-PET unknown primary carcinoma of the head and neck patients often had occult oropharyngeal squamous cell carcinoma, in which unilateral radiation therapy is often safe and recommended. [4,5] Therefore, further investigation is needed regarding treatment volumes within the PET era.

The goal of this study was to evaluate the effects of radiation treatment volume on recurrence and mortality among head and neck cancer patients who underwent PET imaging with unknown primaries and were treated at two different cancer centers.

## Materials and methods

Clinical characteristics of 62 patients with unknown primaries of the head and neck treated from 2000–2015 at two academic medical centers were abstracted from the medical records. All patients had confirmed squamous cell carcinoma. Patients either received radiation therapy to one side of the head and neck (unilateral radiation), or radiation therapy to both sides

of the head and neck (bilateral radiation) (Table 1). In the event of concern for bilateral neck involvement, irradiation to both necks would be an absolute indication regardless of treating center. However, in other instances inclusion of the elective contralateral neck was at the discretion of the treating physician.

All 33 patients treated at Center A received routine PET scan and direct examination in the operating room. All were treated with concurrent radiation therapy (CCRT). Of these, 30 received cisplatin based therapy and 3 were treated with cetuximab. Induction chemotherapy was given prior to CCRT in 4 patients. Twelve patients received a neck dissection either prior to or after CCRT. IMRT was used in 22 patients. The radiation volume always encompassed the ipsilateral oropharynx (base of tongue, tonsil) and level 1B to V lymph nodes. If clinical exam or imaging (including PET) revealed evidence concerning for bilateral neck involvement then both necks, bilateral 1B to V lymph nodes were treated along with the bilateral oropharynx (base of tongue, tonsil). IMRT dosing (70 Gy to gross tumor and oropharynx with 56 Gy to the nodes at risk in 35 fractions) and technique has been previously described [6,7]. 3DCRT was delivered to 70 Gy in 35 fractions to the ipsilateral oropharynx and positive nodes and 50 Gy to the supraclavicular region in 25 fractions.

Patients treated at Center B received routine PET scan and direct examination in the operating room (n = 29). RT was delivered with either three dimensional conformal radiation therapy (3DCRT) (11 patients) or IMRT (18 patients). Target volume delineation and prescription dose was at the discretion of the treating physician based on patient and tumor factors. Sixteen and five patients underwent neck dissection prior to, and after, radiotherapy, respectively. Two patients were treated to the neck without mucosal site irradiation. Three patients were treated only to the oropharynx, 8 patients were treated to oropharynx and nasopharynx, and 15 patients were treated to the orpharynx, nasopharynx, hypopharynx and larynx. Detailed information regarding RT targets was not available in two patients. Nineteen patients received concurrent chemotherapy, consisting of cisplatin in 13, cetuximab in 3, and a combination or other in 3. Median dose to gross disease or highest risk area (assuming neck dissection) was 66 Gy. Median dose to uninvolved but high risk areas was 59.7 Gy (26 patients). Unilateral elective treatment was used in 4 patients, while the rest received bilateral treatment.

## Statistical analysis

Mann Whitney U tests for ordinal data and Chi square and Fischer exact tests for categorical data were conducted to compare demographic and outcome factors between patients treated unilaterally and patients treated bilaterally. Overall survival (OS) and disease free survival (DFS) trends for unilaterally and bilaterally treated patients were estimated using Kaplan-Meier survival curves. The effect of treatment volume on overall survival and disease free survival were examined using multivariate cox proportional hazard regression models. Models were adjusted for age and nodal stage. A p-value of $< 0.05$ was considered statistically significant. All analyses were performed using SAS version 9.4

## Results

The majority of patients were white (88.71%), male (74.19%), and former smokers (54.84%). There were no significant differences in age (p = 0.13), sex (p = 0.65), smoking status (p = 0.17), or N stage (p = 0.61) between patients treated unilaterally and patients treated bilaterally (Table 2).

No significant differences in the frequency of local (p = 0.32), regional (p = 0.50), or distant (p = 0.76) failures were observed between patients treated unilaterally and those treated bilaterally (Table 2). Moreover, Kaplan-Meier estimates for OS (3-year OS bilateral = 71.67%,

**Table 1. Treatment volumes, modalities, and concurrent therapy.**

| | | Center A | Center B |
|---|---|---|---|
| **RT volume** | Unilateral | 28 | 6 |
| | Bilateral | 5 | 23 |
| **RT technique** | IMRT | 22 | 18 |
| | 3DCRT | 11 | 11 |
| **Concurrent chemotherapy** | Cisplatin | 30 | 13 |
| | Cetuximab | 3 | 3 |
| | Other | 0 | 3 |
| | None | 0 | 10 |
| **Surgery** | Yes | 12 | 6 |
| | No | 21 | 23 |

unilateral = 77.90%, p = 0.50, Fig 1) and DFS (3-year DFS bilateral = 77.92%, unilateral = 69.43%, p = 0.63, Fig 2) were similar between the two treatment approaches. Furthermore, there was no statistically significant effect of treatment volume on disease free survival in both univariate (HR = 0.77, 95% CI: 0.28, 2.18, p = 0.63) and multivariate (HR: 0.68, 95% CI: 0.24, 1.93, p = 0.47) analyses (Table 3).

## Discussion

The current study evaluates the approach of two different centers in the management of unknown primary of the head and neck. In this cohort, Center B favored bilateral neck irradiation whereas Center A preferred unilateral coverage. As there were no significant association between nodal stage and radiation volumes, it is likely this difference is driven by preference of the prescribing physician and not by concern for bilateral neck involvement. This study found no significant difference in three year OS or DFS between patients treated with unilateral or bilateral head and neck radiotherapy. In addition, there was no significant difference in rates of local, regional, or distant failures among the treatment groups. Ligey et al [8] also observed no significant difference in tumor control or overall survival between patients treated unilaterally or bilaterally. Similarly, Le at el. reported no oncologic benefit to bilateral neck irradiation as well.[9] Our results are consistent with other published reports [10–13].

While other reports have suggested that bilateral neck radiotherapy may result in improvements in local tumor control and overall survival [14–17], these studies were conducted in an era prior to the more widespread use of PET imaging. A recent meta-analysis of 16 studies concluded that bilateral radiation therapy provided better local and regional control than unilateral radiation therapy [18]. In addition, although not statistically significant, there were trends towards improved overall survival in the bilaterally treated patient population [18]. However, these studies include patient populations from the 1970s to the early 2000s, with the majority of the patients having been treated before the widespread adoption of PET imaging. Even the most recent of these papers, Beldi et al [19] includes a population of patients that ranges from 1998–2004. The majority of patients included in these studies would not have been treated with IMRT[20], but with 2D and 3D radiation therapy.

The studies by Ligey et al (2009) [8], Perkins et al (2012)[10], Lu et al (2009)[11], McMahon et al (2000)[12], Fakhrian et al (2012)[13], and Le et al. (2019)[9] which show similar outcomes regardless of volume irradiated, also include patients treated in prior decades, but had a higher proportion of patients treated in the mid-late 2000s. In addition to more advanced radiation therapy techniques [19], patients treated in more recent years had the benefit of undergoing PET scanning. With the advent of PET scanning, what would have been considered an

**Table 2. Characteristics of unilaterally and bilaterally treated patients with squamous cell carcinoma of the head and neck from an unknown primary (n = 62).**

| Characteristic | Bilateral Treatment (n = 28) Median (range) or N (%) | Unilateral Treatment (n = 34) Median (range) or N (%) | *p-value |
|---|---|---|---|
| **Age (years)** | 57.00 (41–82) | 60.72 (44.65–75.00) | 0.13 |
| **Sex** | | | |
| Male | 20 (71.43) | 26 (76.47) | |
| Female | 8 (28.57) | 8 (23.53) | 0.65 |
| **Smoking Status** | | | |
| Never | 4 (14.29) | 12 (35.29) | |
| Former | 18 (64.29) | 16 (47.06) | |
| Current | 6 (21.43) | 6 (17.65) | 0.17 |
| **Nodal stage** | | | |
| Nodal stage1 | 4 (14.29) | 7 (20.59) | |
| Nodal stage 2 | 15 (53.57) | 20 (57.14) | |
| Nodal stage3 | 9 (32.14) | 7 (20.59) | 0.61 |
| ¶**Local Failure** | | | |
| No | 25 (89.29) | 33 (97.06) | |
| Yes | 3 (10.71) | 1 (2.94) | 0.32 |
| ^**Regional Failure** | | | |
| No | 28 (100.00) | 32 (94.12) | |
| Yes | 0 (0.00) | 2 (5.88) | 0.50 |
| ͻ**Distant Failure** | | | |
| No | 22 (78.57) | 28 (82.35) | |
| Yes | 6 (21.43) | 6 (17.65) | 0.76 |
| **Current Status** | | | |
| Alive | 18 (64.29) | 23 (67.65) | |
| Dead | 10 (35.71) | 11 (32.35) | 0.78 |
| **Disease Status** (excludes deceased) | | | |
| No evidence of disease | 18 (100.00) | 20 (86.96) | |
| Alive with this HN cancer | 0 | 1 (4.35) | 0.50 |
| Alive with other cancer | 0 | 2 (8.70) | |

*Mann Whitney U tests conducted for ordinal data; Chi square or Fisher Exact test conducted for categorical data

¶Local Failure = same site

^Regional Failure = within head neck/surrounding lymph nodes

ͻDistant Failure = metastasis anywhere else in body

unknown primary in past decades is now identified [2,4,5]. This technological innovation narrows the population of patients to true unknown primaries in the modern era and may explain the difference in survival outcomes between older and newer studies.

Local, regional, and distant failures were observed in both treatment groups with no statistically significant differences. The majority of the failures in both groups were at distant sites. Perkins et al (2012) [10] also found equal rates of distant metastasis between the two groups. Distant metastasis is the most common site of failure for unknown primary tumors and represents a substantial risk to a patient's overall survival [2]. As rates of distant metastasis in this study and others do not appear to be altered by unilateral radiation therapy, the benefits of unilateral radiation therapy may be achieved with no significant difference in survival.

The benefits of unilateral radiation therapy over bilateral radiation therapy include decreased toxicity and potentially improved quality of life. Although the results from Reddy

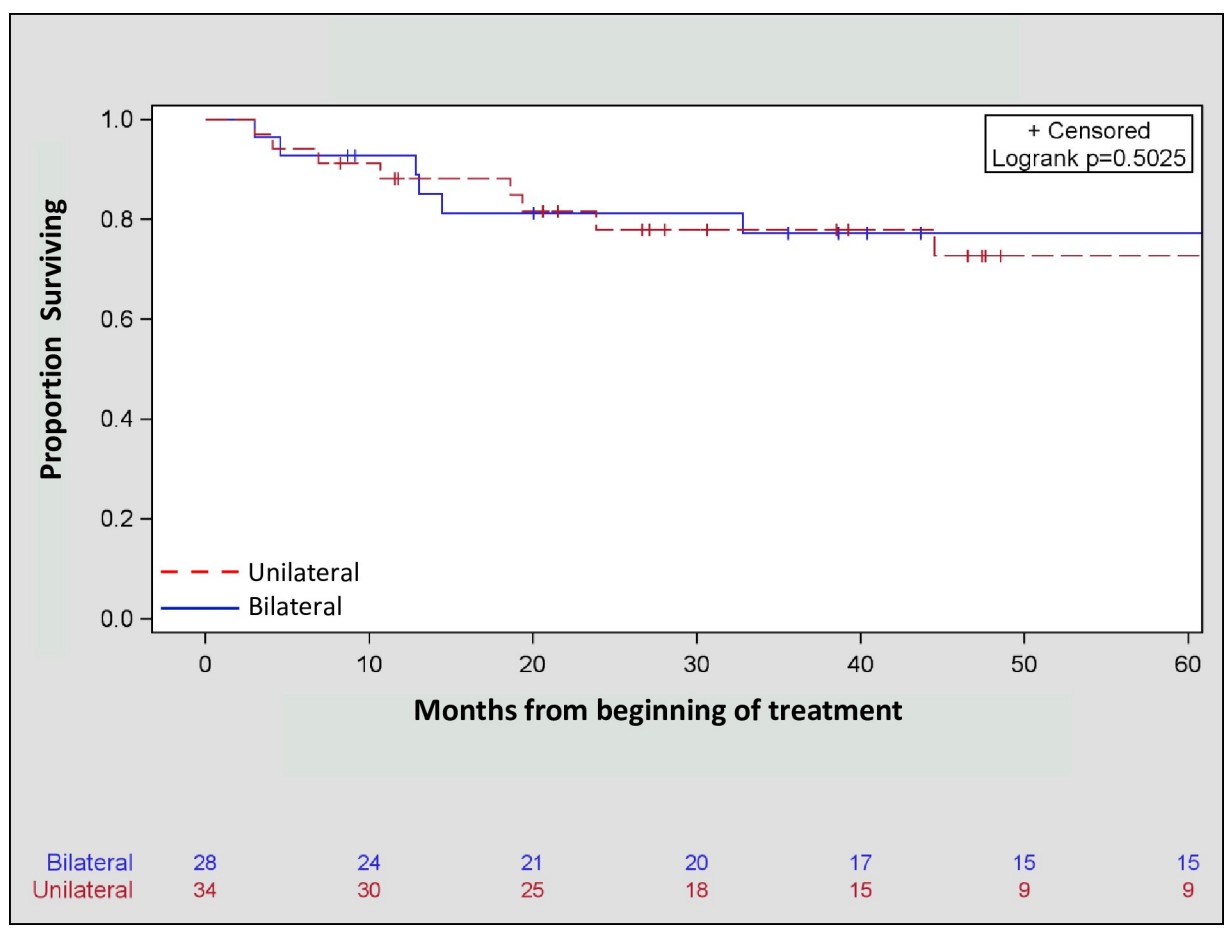

**Fig 1. Kaplan-Meier three-year overall survival of unilaterally (77.90%) and bilaterally (71.67%) treated patients with squamous cell carcinoma of the head and neck with unknown primary (p = 0.50).**

et al (1997) [15] did not favor unilateral radiation therapy, they did observe more mucositis and xerostomia in patients treated with bilateral radiation therapy compared to patients treated with unilateral radiation therapy. Fakhrian et al (2012) [13] also observed more mucositis and xerostomia in bilaterally treated patients than unilaterally treated patients. There were no incidences of severe xerostomia in the unilateral patient group, compared to 4 incidences of severe xerostomia in the bilaterally treated group [13]. Le et al. (2019)[9] reported significantly worse acute dysphagia and mucositis with bilateral neck irradiation, with trends for increased acute laryngeal alteration and xerostomia as well [9]. Formerly, quality of life was not routinely measured making it difficult to assess. Future studies focusing on the beneficial effects of unilateral radiation therapy on quality of life are warranted. In addition to potential quality of life benefits, there is potential for a lower risk of toxicity in the unilateral population should a contralateral recurrence emerge.

This study represents one of the few studies to be conducted on a modern population of patients with squamous cell carcinoma of unknown primary origin of the head and neck. As a result, many of the patients in this study underwent PET imaging and were treated with intensity modulated radiation therapy. Furthermore, this paper is a multi-institutional study minimizing potential bias due to the treatment center.

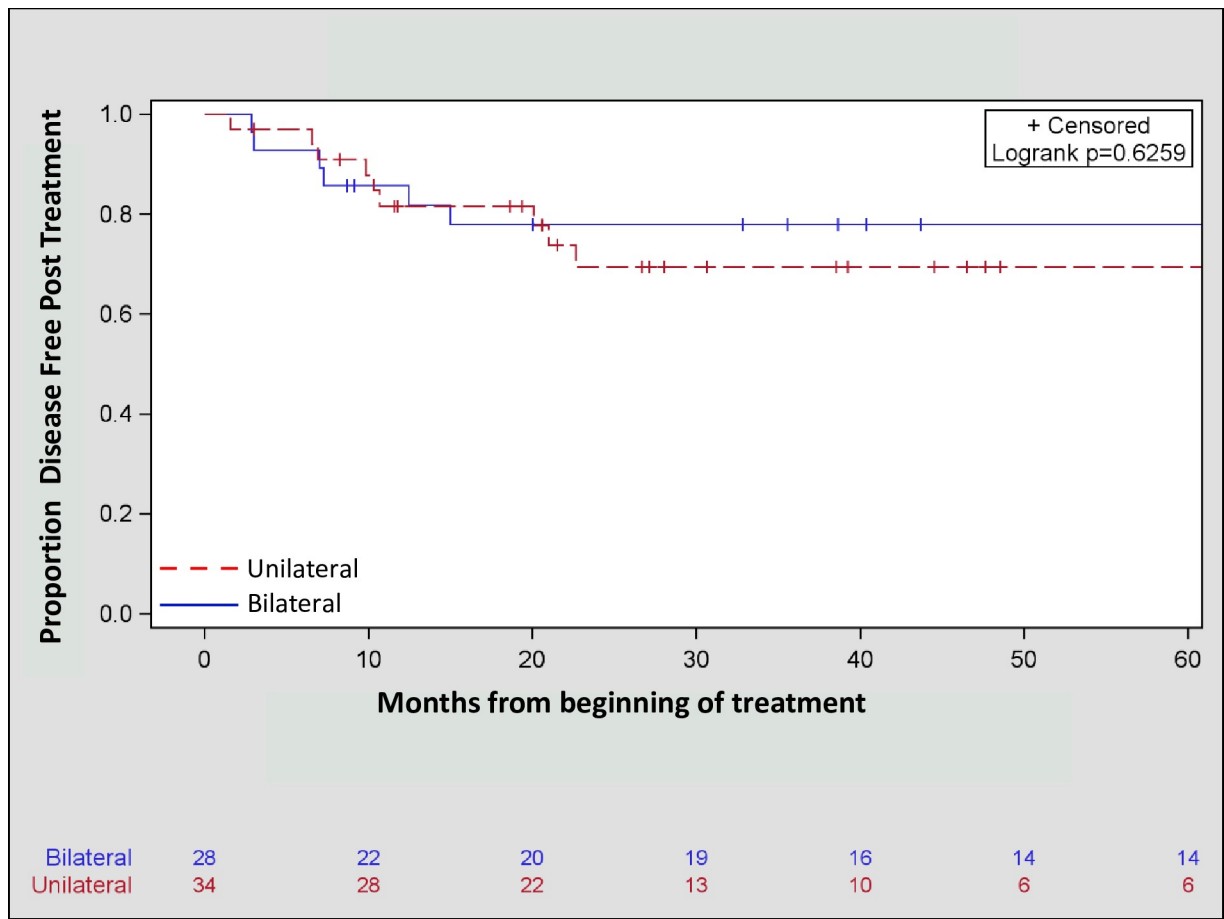

**Fig 2. Kaplan Meier three-year disease free survival of unilaterally (69.43%) and bilaterally (77.92%) treated patients with squamous cell carcinoma of the head and neck from an unknown primary (p = 0.63).**

**Table 3. Cox proportional hazard ratio models for overall and disease free survival in patients with squamous cell carcinoma of the head and neck from an unknown primary (n = 62).**

| Model | Overall Survival | | Disease Specific Survival | |
|---|---|---|---|---|
| | HR (95% CI) | p-value | HR (95% CI) | P-value |
| **Univariate** | | | | |
| Unilateral | 1.00 | | 1.00 | |
| Bilateral | 0.74 (0.30, 1.80) | 0.504 | 0.77 (0.28, 2.18) | 0.627 |
| **Age adjusted** | | | | |
| Unilateral | 1.00 | | 1.00 | |
| Bilateral | 0.82 (0.34, 2.00) | 0.661 | 0.83 (0.29, 2.23) | 0.725 |
| *Multivariate | | | | |
| Unilateral | 1.00 | | 1.00 | |
| Bilateral | 0.74 (0.31, 1.81) | 0.511 | 0.68 (0.24, 1.93) | 0.465 |

Abbreviations: HR, hazard ratio; 95% CI, 95% confidence interval

*Multivariate model adjusted for age and nodal stage

Additionally, examining a modern population such as ours is likely to approximate the current incidence of human papilloma virus (HPV) infection which is known to impact incidence, epidemiology, and outcomes of head and neck cancer [21]. In this HPV era, there may even be differences in outcomes within different subsites of the oropharynx [22]. However, specific data on HPV are lacking in this cohort because most patients were diagnosed by fine needle aspiration (FNA) only and that p16 testing was not widely performed until after 2010. Despite a recent report on the technical feasibility of performing HPV on FNA specimens [23], this was not routine at the time and not done on our specimens.

The entrenched position of many physicians on how to treat unknown primary head and neck cancer has previously caused the aforementioned failure of a cooperative group trial on unilateral versus comprehensive mucosal irradiation. A recent review of the literature from October 2018 is one of the few papers to make recommendations on irradiation volume for unknown primaries.[24] They have recommended avoiding routine irradiation of all lymph node levels. Instead, they favor unilateral radiation for those with low risk N stages (N1, N2A, N2B) and consider bilateral irradiation for those with risk of contralateral metastases (N2C, N3).Furthermore, they found no survival benefit of bilateral radiation therapy accompanied by increased toxicity.[24] Of the three co-authors from center B of this publication, two of the co-authors now consider volume directed ipsilateral irradiation of the oropharynx and neck in the majority of patients and the third co-author expresses a hesitancy to change existing practice prior to a demonstration of diminished toxicity from unilateral therapy.

## Conclusions

Results of this study show that unilateral radiation therapy for unknown primaries of the head and neck does not have inferior overall survival, disease free survival, or increased rates of failures compared to bilateral radiation therapy. No significant differences in survival combined with the potential benefits of improved quality of life and decreased toxicity make unilateral radiation therapy an attractive prospect.

## Author Contributions

**Conceptualization:** Michael Mix, Varun Chowdhry, Michael A. Lacombe, Jeffrey A. Bogart, Austin Iovoli, Hassan Arshad, Kimberly Wooten, Vishal Gupta, Wesley L. Hicks Jr, Seung S. Hahn, Anurag K. Singh.

**Data curation:** Alexis Platek, Michael Mix, Varun Chowdhry, Mark Farrugia, Michael A. Lacombe, Jeffrey A. Bogart, Luke Degraaf, Austin Iovoli, Hassan Arshad, Kimberly Wooten, Vishal Gupta, Wesley L. Hicks Jr, Seung S. Hahn.

**Formal analysis:** Alexis Platek, Mary E. Platek.

**Writing – original draft:** Alexis Platek, Luke Degraaf, Mary E. Platek, Anurag K. Singh.

**Writing – review & editing:** Alexis Platek, Mark Farrugia, Anurag K. Singh.

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
