## [Decision Letter · Decision Letter 0]

12 Feb 2020

PONE-D-19-35352

Unilateral irradiation for unknown primaries of the head and neck in the era of FDG-PET

PLOS ONE

Dear Dr. Singh,

Thank you for submitting your manuscript to PLOS ONE. After careful consideration, we feel that it has merit but does not fully meet PLOS ONE’s publication criteria as it currently stands. Therefore, we invite you to submit a revised version of the manuscript that addresses the points raised during the review process.

We would appreciate receiving your revised manuscript by Mar 28 2020 11:59PM. To enhance the reproducibility of your results, we recommend that if applicable you deposit your laboratory protocols in protocols.io, where a protocol can be assigned its own identifier (DOI) such that it can be cited independently in the future. For instructions see: http://journals.plos.org/plosone/s/submission-guidelines#loc-laboratory-protocols

We look forward to receiving your revised manuscript.

Kind regards,

Jason Chia-Hsun Hsieh, M.D. Ph.D

Academic Editor

PLOS ONE

Additional Editor Comments (if provided):

Most of the questions were answered adequately; however, a minor revision is still required.

Journal Requirements:

Reviewers' comments:

Reviewer's Responses to Questions

**Comments to the Author**

1. Is the manuscript technically sound, and do the data support the conclusions?

Reviewer #1: Yes

Reviewer #2: Partly

Reviewer #3: Yes

Reviewer #4: Partly

2. Has the statistical analysis been performed appropriately and rigorously? 

Reviewer #1: Yes

Reviewer #2: Yes

Reviewer #3: Yes

Reviewer #4: Yes

3. Have the authors made all data underlying the findings in their manuscript fully available?

Reviewer #1: Yes

Reviewer #2: Yes

Reviewer #3: Yes

Reviewer #4: Yes

4. Is the manuscript presented in an intelligible fashion and written in standard English?

Reviewer #1: Yes

Reviewer #2: Yes

Reviewer #3: Yes

Reviewer #4: Yes

5. Review Comments to the Author

Reviewer #1: The authors have addressed previous raised comments and concerns. The manuscript has overall improved. The novelty has been clearly explained and discussed. I believe the manuscript can be published as it stands right now.

Reviewer #2: I agree with the original comments from review 1 and 2. Generally, the limitations of the study were not well-discussed. First, although the patient cohort comes from two centers but the study still only have 62 patients which is the major weakness. Other confounding factors raised by the reviewers were not addressed in the discussion section.

Reviewer #3: This study is the evaluation of recurrence and mortality according to the different radiation volume among head and neck cancer patients who underwent PET imaging with unknown primaries and were treated at two different cancer centers.

The authors concluded that that unilateral radiation therapy for unknown primaries of the head and neck does not have inferior overall survival, disease free survival, or increased rates of failures compared to bilateral radiation therapy. No significant differences in survival combined with the potential benefits of improved quality of life and decreased toxicity make unilateral radiation therapy an attractive prospect.

There is no description of the toxicity according to the different radiation therapy volume. Is there any analysis of the quality of life or side effects?

Reviewer #4: The authors present an interesting study to evaluate the whether unilateral or bilateral irradiation should be considered in head and neck cancer patients. From their analysis, they did not produce any conclusive or statistically significant results to conclude that one method should be used more than another. The work seems to be an underpowered and slightly unconvincing study without more patients or methods to assess toxicity. Nonetheless, it may be important to note that there were no significant differences between the two cases and that the unilateral case does not result in worse outcomes. The paper would be strengthened if the authors address the following issues:

--The title of the paper remains too focused on unilateral irradiation when that is not necessarily warranted from their results. A more appropriate title should be given.

--PET was still not appropriately introduced in the abstract as there was only a vague mention of the era of PET. There is no explanation as to why PET is important or what it can be used for. An additional sentence is necessary to make this connection.

--The unilateral vs. bilateral case mismatch in each center needs to be addressed and explained in 1st paragraph of the Materials and Methods section (e.g. why did most patients receive bilateral treatment in Center B? Was this due to evidence of bilateral neck involvement as in Center A?).

--The sentence starting on line 203 is misleading (“As in our study…accompanied by increased toxicity.”) as the authors did not evaluate toxicity in their own study. This sentence should be updated to clarify this point.

6. PLOS authors have the option to publish the peer review history of their article (what does this mean?). If published, this will include your full peer review and any attached files.

Reviewer #1: No

Reviewer #2: No

Reviewer #3: No

Reviewer #4: No

---

## [Author Response · Author response to Decision Letter 0]

3 Mar 2020

Reviewer #1: The authors have addressed previous raised comments and concerns. The manuscript has overall improved. The novelty has been clearly explained and discussed. I believe the manuscript can be published as it stands right now.

Reviewer #2: I agree with the original comments from review 1 and 2. Generally, the limitations of the study were not well-discussed. First, although the patient cohort comes from two centers but the study still only have 62 patients which is the major weakness. Other confounding factors raised by the reviewers were not addressed in the discussion section.

Reviewer #3: This study is the evaluation of recurrence and mortality according to the different radiation volume among head and neck cancer patients who underwent PET imaging with unknown primaries and were treated at two different cancer centers.

The authors concluded that that unilateral radiation therapy for unknown primaries of the head and neck does not have inferior overall survival, disease free survival, or increased rates of failures compared to bilateral radiation therapy. No significant differences in survival combined with the potential benefits of improved quality of life and decreased toxicity make unilateral radiation therapy an attractive prospect.

There is no description of the toxicity according to the different radiation therapy volume. Is there any analysis of the quality of life or side effects?

Thank you for this comment. This study did not include quality of life/toxicity as this data was not routinely collected at the participating centers during the study period however we have referenced and discussed previous studies which investigated these factors in lines 181-194. 

Reviewer #4: The authors present an interesting study to evaluate the whether unilateral or bilateral irradiation should be considered in head and neck cancer patients. From their analysis, they did not produce any conclusive or statistically significant results to conclude that one method should be used more than another. The work seems to be an underpowered and slightly unconvincing study without more patients or methods to assess toxicity. Nonetheless, it may be important to note that there were no significant differences between the two cases and that the unilateral case does not result in worse outcomes. The paper would be strengthened if the authors address the following issues:

--The title of the paper remains too focused on unilateral irradiation when that is not necessarily warranted from their results. A more appropriate title should be given.

Thank you for this comment. We have changed the title to “Evaluation of radiation treatment volumes for unknown primaries of the head and neck in the era of FDG PET”

--PET was still not appropriately introduced in the abstract as there was only a vague mention of the era of PET. There is no explanation as to why PET is important or what it can be used for. An additional sentence is necessary to make this connection.

Thank you for this suggestion. We have rewritten a portion of the abstract objectives seen on lines 29-32. “Positron-emission tomography (PET) has improved identification of the primary tumor as well as occult nodal burden in cancer of the head and neck. Nevertheless, there are still patients where the primary tumor cannot be located. In these situations, the standard of care is comprehensive head and neck radiation therapy however it is unclear whether this is necessary.”

--The unilateral vs. bilateral case mismatch in each center needs to be addressed and explained in 1st paragraph of the Materials and Methods section (e.g. why did most patients receive bilateral treatment in Center B? Was this due to evidence of bilateral neck involvement as in Center A?).

Thank you for this comment. This mismatch lies in approach of the treating physician, an approach that is grounded in philosophy rather randomized data given the lack of prospective trials examining this fairly rare clinical scenario. We had addressed what absolute indications for bilateral neck irradiation are in the text (lines 91-94). Furthermore, we have highlighted the fact there were no association between nodal stage and radiation volumes and clarified that the driving force between bilateral vs unilateral coverage was the decision to electively cover the contralateral neck in the absence of PET avid disease. (lines 143-147) This decision is largely a difference in treatment philosophy, which differed across the two centers. We then discuss how these data potentially impact this treatment approach of the involved physicians in lines 215-219. 

--The sentence starting on line 203 is misleading (“As in our study…accompanied by increased toxicity.”) as the authors did not evaluate toxicity in their own study. This sentence should be updated to clarify this point. 

Thank you for this comment. This has been addressed (line 215)

---

## [Editor Report · Decision Letter 1]

16 Mar 2020

Evaluation of radiation treatment volumes for unknown primaries of the head and neck in the era of FDG PET

PONE-D-19-35352R1

Dear Dr. Singh,

We are pleased to inform you that your manuscript has been judged scientifically suitable for publication and will be formally accepted for publication once it complies with all outstanding technical requirements.

With kind regards,

Jason Chia-Hsun Hsieh, M.D. Ph.D

Academic Editor

PLOS ONE

Additional Editor Comments (optional):

The topic is well-presented and interesting in the head and neck cancer field. After revision, all the questions were addressed this time.
---

## [Editor Report · Acceptance letter]

18 Mar 2020

PONE-D-19-35352R1 

Evaluation of radiation treatment volumes for unknown primaries of the head and neck in the era of FDG PET 

Dear Dr. Singh:

I am pleased to inform you that your manuscript has been deemed suitable for publication in PLOS ONE. Congratulations! Your manuscript is now with our production department. 

With kind regards,

on behalf of

Dr. Jason Chia-Hsun Hsieh 

Academic Editor

PLOS ONE